# How Tree Decline Varies the Anatomical Features in *Quercus brantii*

**DOI:** 10.3390/plants12020377

**Published:** 2023-01-13

**Authors:** Forough Soheili, Hazandy Abdul-Hamid, Isaac Almasi, Mehdi Heydari, Afsaneh Tongo, Stephen Woodward, Hamid Reza Naji

**Affiliations:** 1Department of Forest Sciences, Ilam University, Ilam 67187-73654, Iran; 2Faculty of Forestry and Environment, Universiti Putra Malaysia, Serdang 43400, Selangor, Malaysia; 3Faculty of Science, Department of Statistics, Razi University, Kermanshah 67144-14971, Iran; 4Department of Forest Science and Engineering, Sari University of Agricultural Sciences and Natural Resources, Sari 48181-68984, Iran; 5School of Biological Sciences, University of Aberdeen, Aberdeen AB24 3UU, UK

**Keywords:** drought, tree decline, *Quercus brantii*, cellular characteristics, calcium oxalate crystals

## Abstract

Drought has serious effects on forests, especially semi-arid and arid forests, around the world. Zagros Forest in Iran has been severely affected by drought, which has led to the decline of the most common tree species, Persian oak (*Quercus brantii*). The objective of this study was to determine the effects of drought on the anatomical structure of Persian oak. Three healthy and three declined trees were sampled from each of two forest sites in Ilam Forest. Discs were cut at breast height, and three sapwood blocks were taken near the bark of each tree for sectioning. The anatomical characteristics measured included fiber length (FL), fiber wall thickness (FWT), number of axial parenchymal cells (NPC), ray number (RN), ray width (RW), and number of calcium oxalate crystals. Differences between healthy and declined trees were observed in the abundance of NPC and in RN, FL, and FWT, while no differences occurred in the number of oxalate crystals. The decline had uncertain effects on the FL of trees from sites A and B, which showed values of 700.5 and 837.3 μm compared with 592.7 and 919.6 μm in healthy trees. However, the decline resulted in an increase in the FWT of trees from sites A and B (9.33 and 11.53 μm) compared with healthy trees (5.23 and 9.56 μm). NPC, RN, and RW also increased in declined individuals from sites A and B (28.40 and 28.40 mm^−1^; 41.06 and 48.60 mm^−1^; 18.60 and 23.20 μm, respectively) compared with healthy trees (20.50 and 19.63 mm^−2^; 31.60 and 28.30 mm^−2^; 17.93 and 15.30 μm, respectively). Thus, drought caused measurable changes in the anatomical characteristics of declined trees compared with healthy trees.

## 1. Introduction

Ongoing global warming is increasing stress in most of the Earth’s ecosystems [1]. As the duration, frequency, and intensity of droughts increase, forest productivity decreases and tree mortality rates are expected to increase [2,3].

In forest ecosystems, trees are exposed to a combination of biotic and abiotic stressors that limit their growth potential [4]. Abiotic stressors such as drought weaken tree response to environmental conditions, leading to further damage and, in extreme cases, death [5]. In the last 20–30 years, the effects of climate change have led to an increase in damage to forest ecosystems, reducing the adaptive capacity of plant communities in these ecosystems [6]. Forest decline is a complex destructive phenomenon that leads to declining tree growth and, over time, tree death [7]. The decline is often a result of extreme climatic events, which can also lead to further damage caused by biotic factors [8].

In recent decades, the decline of oak species has been observed throughout the Mediterranean region and is clearly related to dry seasons [9]. Zagros Forest, a semi-arid region in western Iran, is the largest oak forest in the world with an area of more than 5 million hectares, dominated by Persian oak (*Quercus brantii* L.) [10]. In this forest, oaks help to store and regulate water resources, conserve soil, mitigate climate change, and improve the socioeconomic conditions of human communities [11]. This forest is rich in biodiversity and provides habitat for numerous species of organisms. Since 2000, there have been recurrent droughts in Iran, especially in Zagros Forest, which have led to the disturbing death of trees, mainly Persian oaks. Drought is considered to be the most important influencing factor, followed by reduction in photosynthetic rate, reduction in annual growth increment, weakening of trees, and invasion of fungi and wood beetles [12].

Correlating variations in wood structure and tree defense strategies could help to better understand plant performance, especially in the face of climate change [13,14]. The plant cell wall, a complex macromolecular structure that surrounds and protects the cell contents, is essential to plant survival. Under drought conditions, negative stresses increasingly occur in the xylem due to water stress, causing the thickness of nearby fiber walls to prevent the vessels from imploding under the altered tension [15]. The fluctuations in climatic conditions affect trees and lead to permanent records in the xylem tissue. Therefore, trees can easily be considered as reliable natural documents that reveal past environmental events [16,17]. The xylem parenchymal cells are important for facilitating water conduction, serving as a water reservoir to prevent embolisms and providing osmotic agents to aid in the repair of embolisms [18]. The functions of the parenchyma (axial and ray cells) are (a) storage of water, nonstructural carbohydrates, and mineral elements [19,20,21]; (b) defense system against decay fungi and pathogens [22]; (c) aid in the transition from sapwood to heartwood [23]; and (d) biomechanical contributor, especially the ray parenchyma [24]. 

Calcium oxalate crystals form in the cells of many plant species, both monocotyledonous and dicotyledonous plants [25], and their functions include balancing the ionic balance, protecting and defending the plant, and detoxifying potentially harmful chemicals [26,27,28]. Serdar and Demiray [29] reported an abundance of intracellular crystals with different shapes in three different oaks from Turkey.

Despite the key role of the parenchyma in tree functioning, there is little research on the changes that occur in wood parenchymal cells in response to potential factors that affect their abundance. Morris et al. [30] found that an increase in the average number of parenchymal cells in the xylem occurred as a result of the increase in the annual air temperature. The effect was more pronounced in regions with average annual temperatures above 16 °C. In cold regions, where the total number of parenchymal cells in wood is lower than in the tropics, the role of the parenchyma as a defensive tissue may have a higher priority than other functions [31]. 

In recent decades, Persian oak forests have shown symptoms of a new phenomenon known as oak decline, most likely caused by drought stress [32]. Understanding the responses of trees to drought could be considered an interesting and active discipline of plant science research [33].

In the absence of specific information on changes in the cell anatomical structure of wood fibers and axial and ray parenchymal cells during drought stress, the work described in this article could provide valuable insights into this aspect of tree–environment interactions. Because all components of wood tissue may respond differently to environmental stress, we hypothesized the role of wood parenchymal cells and the changes in fiber and cell crystal properties during tree response to prolonged drought.

## 2. Results

### 2.1. Fiber Biometric Characterization

#### 2.1.1. Fiber Length (FL)

The differences in FL in trees from the two sites and site×decline interactions were significant (*p* < 0.01) in sapwood samples collected near the bark (Table 1). Regarding site A, FL was significantly greater in sapwood near the bark in declined trees (700.5 μm) than in healthy individuals (592.7 μm). In sapwood tissue samples from site B, FL was significantly shorter in dying trees (837.3 μm) than in healthy trees (*p* < 0.05) (919.6 μm; Figure 1A).

#### 2.1.2. Fiber Wall Thickness (FWT)

The effects of decline, site, and their interaction on FWT were significant (*p* < 0.01) in sapwood adjacent to the bark (Table 1). FWT was significantly higher (*p* < 0.05) in declined trees from sites A and B (9.33 μm and 11.53 μm, respectively) than in healthy trees from the two sites (5.23 μm and 9.56 μm; Figure 1B).

#### 2.1.3. Number of Axial Parenchymal Cells (NPC)

Decline had significant effects on the NPC (*p* < 0.01) in sapwood near the bark (Table 1). The NPC in fallen trees from sites A and B (28.4 and 28.4 mm^−2^, respectively) was higher than that in healthy trees (20.5 and 19.6 mm^−2^; Figure 1C). 

#### 2.1.4. Ray Number (RN)

The effects of decline and site×decline interaction on the RN were significant in sapwood near the bark (*p* < 0.01; Table 1). The RN was significantly higher in the sapwood of declined trees (41.06 and 48.60 mm^−2^) than in healthy trees (31.60 and 28.30 mm^−2^) from the two sites (Figure 1D).

#### 2.1.5. Ray Width (RW)

RW in sapwood was significantly affected by decline (*p* < 0.01); interactions between decline and site also showed a significant effect on RW (*p* < 0.01; Table 1). RW was significantly higher (*p* < 0.01) in declined trees (23.20 μm) than in healthy trees (15.30 μm) in sapwood near the bark of trees from site B, but no significant differences were observed between declined and healthy trees from site A (Figure 1E). 

#### 2.1.6. Number of Crystals (NC)

Crystals were only observed in ray parenchymal cells. Differences in the NC between sites were significant (*p* < 0.05) in sapwood near the bark (Table 1). The NC in declined wood was higher in healthy trees (*p* < 0.05) from the two sites (Figure 2). In addition, the highest NC was observed in a sample from site A (Figure 1F). 

#### 2.1.7. Principal Component Analysis (PCA)

The first and second axes of the PCA explained approximately 34.7 and 28.0% of the variance in sapwood from healthy and declined *Quercus brantii* affected by drought stress at the two sites. For declined oak trees in Dareh-shahr, the most important variables were FWT, RW, RN, and NPC, and the most important trait of healthy trees from the Ilam site was FL. Similarly, in declined trees from the Ilam site, the most important trait was the NC. It was investigated that drought-affected oaks from both climatic regions could be distinguished from healthy trees based on the anatomical characteristics. The RW, RN, and NPC traits were significantly more distinct and common in declined oak trees than in healthy trees from both the Dareh-shahr and Ilam sites (Figure 3; Table 2).

## 3. Discussion

Drought, as a severe environmental condition, affects tree growth [34]. To reduce water loss and adapt to drought, some typical changes in physiological processes occur, such as stomatal closure, reduction in photosynthetic assimilation, and cessation of leaf and shoot growth [35]. In addition, Soheili et al. [32] found that the number of earlywood vessels in Persian oaks decreased due to tree decline. These conditions greatly reduce cambial activity and result in strong inhibition of radial growth. This reduction in tree growth is known to be a result of reduced auxin levels [36]. As a result, the growth of the tree stem diameter decreases. As noted by Nola et al. [37], among angiosperms, the genus *Quercus* is more sensitive to drought than other species. This suggests that they may be less competitive under drier conditions in the future.

The influence of drought stress on changes in the anatomical characteristics of *Quercus brantii* wood was studied to determine the effects on living parenchymal cells in sapwood, a subject that has received little attention in the past. The results of this work provide a more solid basis for establishing a functional hypothesis about the role of the parenchyma in wood and the changes in fiber and cell crystal properties during tree response to drought.

Cell wall architecture is important for stress resistance in plants [38,39]. Cell wall synthesis is a flexible component of plant anatomy that can be altered to better cope with the effects of fluctuating biotic and abiotic factors [40]. In this work, according to the FL and FWT values, fibers in sapwood near the bark were shorter and thicker, respectively, in declined trees than in healthy trees (Figure 1A,B and Figure 3). This could lead to higher wood density in declined Persian oaks, as reported in our last publication (Soheili et al. [32]). Previous publications [41,42] suggested that smaller anatomical structures (small- diameter vessels and thinner cell walls) in groups of weakened and dead ash trees (*Fraxinus spp*.) and *Quercus brantii* were due to low activity during the differentiation of the cambium into wood cells. The diameter of earlywood vessels and the area of earlywood vessels were also reduced in Persian oak affected by decline [32]. In addition, Hacke [43] and Wildhagen et al. [44] found that new xylem cells with thicker walls formed under drought compared with unaffected wood. The thickening of cell walls under drought is an important response to improve cell stability and prevent their collapse when pressure on the hydraulic system increases [45].

Environmental stresses such as drought during wood formation affect the size of cells formed subsequently by affecting the rate of cell division and expansion [46]. These physiological changes can sometimes help the plant to adapt to new conditions [47,48]. Naji and Taher Pour [49] stated that abiotic stress, such as a dust storm, can cause significant changes in wood structure, especially fiber properties, in seedlings of Persian oak. Based on the hypothesis that environmental stress can disrupt the hormonal regulation of wood formation as well as the tree’s ability to conduct water [50], it was hypothesized that environmental stress during wood formation leads to structural changes and thus changes in plant properties.

Living parenchymal cells found in the secondary xylem of woody plants can provide a dynamic response to xylem infection and mechanical damage. Furthermore, their role is critical to our understanding of tree defense mechanisms [51]. The xylem parenchyma is known to be a storage compartment in cells [52]. In addition, cells of the ray parenchyma are an important component for the radial water transport between phloem and xylem and for the storage of water, carbohydrates, and other nutrients [53]. Our results suggest that stem ray parenchymal cells respond to long-term environmental conditions such as climate change [54]. Parenchymal cell count (axial and ray) and ray width were greater in declined trees than in healthy trees from the two sites (Figure 1C–E and Figure 3), consistently with the results of previous work on Ulmaceae [55,56,57]. von Arx et al. [54] found that the abundance of ray parenchymal cells in wood tissue may reflect tree vigor and defense against stress. They are able to transport more nutrients to strengthen the tree.

The role played by parenchymal cells in relation to tree hydraulics and their involvement in tree defense strategy against pathogens is not clearly understood, especially since the two responses are interrelated. Parenchymal cells can indirectly prevent the invasion of decay fungi, for example, by separating air-filled vessels. The associated woody cells can act as a buffer zone by refilling emboli in the vessels or creating a continuous flow of water in the otherwise compromised xylem [58,59,60]. Tyloses, as balloon-like swellings that usually form when adjacent parenchymal cells invade dead vascular cells, are a consequence of biotic/abiotic stress in trees, especially in ring-porous species with wide vessels such as oaks [61]. This phenomenon was greatly increased in declined Persian oaks when they were affected by drought [32]. 

Another possible reason for the increase in the average number of axial parenchymal and ray parenchymal cells in the woody tissues of declined trees could be related to the accumulation of antimicrobial compounds such as phytoalexins or phenolic compounds and the deposition of suberin to prevent the spread of fungi [62]. 

Calcium oxalate crystals (CaOx) are widely distributed in flowering plants, indicating an important role in growth and development processes. It should be noted that Table 1 and Figure 1F show no significant effects of decline on the frequency of crystals in the sapwood of healthy and declined trees. Figure 2 shows the frequency of crystals in the ray parenchyma cells of declined trees from two different sampling sites. An abundance of crystals was also observed in biotically stressed trees. Previous research has shown that the deposition of calcium oxalate crystals is high in the parenchymal cells of trees affected by biotic stress such as wood-destroying fungi [63,64]. Figure 1F and Figure 2 show that the differences in the NC were highly related to the site and not to the effects of drought. Therefore, the causes of the differences in the NC between the different sites could have been due to the differences in the elemental mineral content of the soil or in the soil chemistry [65]. 

## 4. Conclusions

Xylem traits help to better understand and predict woody plant responses to climate change on local and global scales. They may also provide an answer to some questions about tree response and performance to changing environmental conditions. Drought stress has major destructive effects on Persian oak (*Quercus brantii*) in Zagros Forest in Iran. Therefore, the work presented here investigated the changes in wood anatomy resulting from drought stress. Previous work on drought-stressed *Q. brantii* has not quantified the effects of this abiotic problem on anatomical characteristics. In general, a decrease in fiber length and an increase in fiber wall thickness were observed in declined wood near the bark. In addition, the number of axial parenchymal and ray parenchymal cells, and the width of the rays were greater in the sapwood of samples from declined trees than in healthy trees. A large number of crystals were present in the ray parenchyma of sapwood sampled immediately adjacent to the bark, although there were no quantitative differences between declined and healthy trees. Further research combining physiology, morphology, and molecular techniques is needed to better understand the role of the variables studied in this work on tree defense and to evaluate the possible role in resistance to biotic and abiotic stresses. In addition, these results can be used as basic information on the effect of drought stress on anatomical changes in *Q. brantii*, and tree responses and adaptations to environmental stress in the future.

## 5. Material and Methods

### 5.1. Study Areas

This study was conducted on Persian oak trees (*Quercus*
*brantii*), both healthy individuals and those affected by drought stress, from two forest sites: Dareh-shahr and Ilam in Zagros Forest, Ilam Province, western Iran (Figure 4). The forest sites are dominated by *Q. brantii* [66]. The main characteristics of the study sites, and the precipitation and temperature data from the nearest meteorological station [67] are tabulated in Table 3. Furthermore, the warmest and coldest months at all three sites are August and January, respectively. The climate classification based on the De Martonne Aridity Index (IDM) is also shown in Table 3.

The severity and duration of drought stress were determined using the Standardized Precipitation Index (SPI), as shown in Figure 5. According to Figure 5, the drought began in 2000, and the index was negative and below normal in most years, indicating moderate-to-severe drought.

### 5.2. Sampling Method

Due to the protected status of Zagros Forest, the number of trees we could sample from each stand was limited. For the declined trees, sampled discs were taken from the trunks of some trees that had been cut by the Department of Natural Resources of Ilam Province as part of a plan to remove damaged trees from the forest. Three healthy and three completely declined trees were sampled from each stand, i.e., the declined and healthy trees were cut from the same stand. A total of 12 trees with diameters at breast height (DBHs) of 30–40 cm were selected from two stands (see Table 3). To avoid the comparison of individual trees of different sizes and ages, trees were taken from a narrow range of DBHs. A disc approximately five-centimeter thick was taken from each fallen tree at breast height for further study. The discs from declined trees were cut with a chainsaw and then sanded using a flat sanding machine (Makita BO4901). More information on the sampling method can be found in our published paper in the journal *Forests* [32]. The study areas and materials of the current work were similar to those in the published work mentioned above, but the difference between the two was in the number of trees selected for the study. In the study by Soheili et al. [32], we measured some microstructures such as tree ring width, vessel characteristics, and wood density in declined trees to compare the variations in the radial direction from bark to pith; therefore, only six declined trees were selected for the study. However, in the present study, three healthy and three declined trees were chosen from each site for the measurement of the variables explained here. Since this is a protected forest and we did not have permission to fell healthy trees, some core samples were taken with an increment borer (HAGLÖF SWEDEN^®®^) from each tree at breast height.

### 5.3. Sample Preparation for Microscopic Investigation

Three small blocks of sapwood attached to the bark, approximately 1 × 1 × 1 cm in size, were cut radially from each disc. Each sample block contained nearly 3–4 annual growth rings. The blocks were labelled with the tree number, location, and class of decline and were fixed in classic fixative FAA according to Ruzin [69] (formaldehyde: ethanol alcohol 95%: Acetic acid; 10%:50%:5% + 35% doubled distilled water). Immediately prior to sectioning, the blocks were softened by heating to 60 °C for 24 h in an oven (Tehran STR).

### 5.4. Wood Sample Processing and Variable Measurement

Wood sections were prepared with no embedding according to the standard protocols by Gartner and Schweingruber [70]. Tangential sections with a thickness of 15–20 μm were prepared using a rotary microtome (POOYAN MK 1110, Binalood, Iran). Due to the strong influence of drought on the trees, the growth rings were narrow; therefore, both earlywood and latewood were included in the tangential sections. Sections were stained with 0.1% (*w/v*) aqueous safranin (Safranin O; Sigma Aldrich, England) for 5–10 min, rinsed three times in distilled water, and dehydrated with an ethanol series of 60%, 85%, 95%, and absolute for approximately 15 min each. The dehydrated sections were fixed on glass slides with Canada balsam [71] and examined under a light microscope (Olympus, CX22LED; Japan) at 100× and 400× magnifications. The report of magnifications in the text are based on the total magnification of ocular and objective lenses. Images were taken with a digital camera connected to a computer. Measurements were performed using Mosaiv ver. 2.0 software (True Chrome metrics, China). The number of axial parenchymal cells (NPC mm^−1^), the number of rays (RN mm^−1^), ray width (RW μm), and the number of crystals (NC mm^−2^) were measured following the IAWA committee (1989) guidelines in both healthy and declined trees at a magnification of 100×. Crystals were distributed in both ray parenchymal cells and axial parenchymal cells. The number of crystals varied in the cells, as most cells contained no crystals, while some cells had at least one and more. In addition, we considered the number of crystals for counting, and the measurement was performed in fields with an area of 1 mm^2^. The fields for measuring the crystals in the parenchymal cells were centered on the locations of sections with no care. This was repeated in some fields to cover all parts of the section [72]. Due to the spindle shape of the ray parenchymal cells, the width was measured in the middle part of the rays. The axial and ray parenchymal cells were measured individually, with 60 counts for each site.

To measure fiber length (FL) and fiber wall thickness (FWT), matchstick-sized pieces of wood were removed from each block with a sharp blade and macerated in a solution of 1.5 g of sodium chlorite in 25 mL of distilled water with 8 drops of glacial acetic acid. The mixture was then placed in a bain-marie water bath at 80 °C for approximately 24 h. The suspension was carefully rinsed several times in distilled water before five drops of Safranin O were added. The free wood fibers were mounted on slides in 30% glycerol and then examined with an Olympus microscope at 100× and 400× magnifications to measure FL and FWT, respectively. Tip-to-tip lengths of undamaged, intact fibers were digitized for measurement. In total, measurements were based on 60 cells in 10 standard fields with an observation area of 1 mm^2^ for each site.

### 5.5. Statistical Analysis

First, we performed the Shapiro–Wilk and Levene’s tests to determine whether using ANOVA or non-parametric equivalents. The results showed that the dependent variables were normally distributed and that the variances were homogenous. To examine the separate and combined effects of the factors of the two sites (A and B) and the severity of decline (healthy and declined trees) on the dependent variables, we used two-way ANOVA and Duncan’s multiple comparison test methods. The means and standard errors of wood anatomical characteristics between declined and healthy trees from the two sites were calculated. The SPSS ver. 21 statistical software package was used for all statistical analyses. Principal component analysis (PCA) based on the correlation matrix using PC -Ord version 5.0 was used to examine multivariate correlations (i.e., relationships among wood anatomical characteristic traits of declined and healthy trees from the two sites).

## Figures and Tables

**Figure 1 plants-12-00377-f001:**
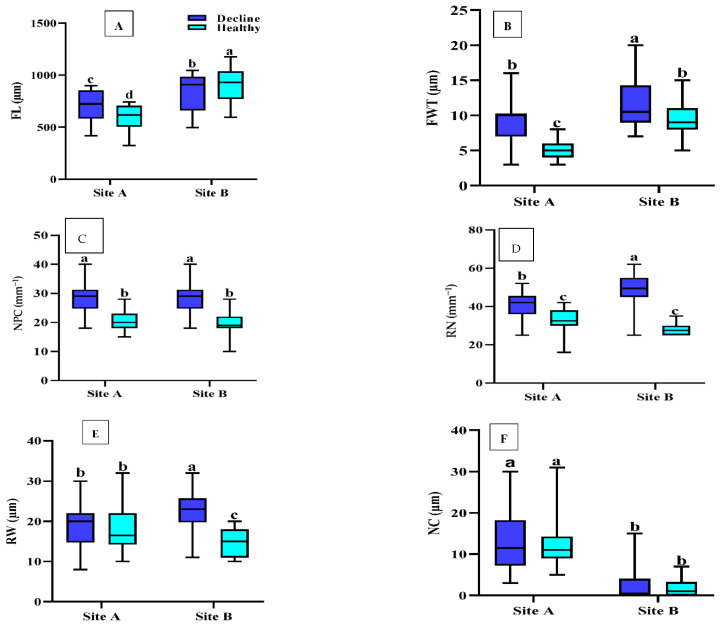
Boxplots of wood anatomy (**A**) FL, fiber length; (**B**) FWT, fiber wall thickness; (**C**) NPC, number of axial parenchymal cells; (**D**) RN, ray number; (**E**) RW, ray width; (**F**) NC, number of crystals) of sapwood from declined and healthy *Quercus brantii* from two sites in Iran. Site A, Dareh-shahr; site B, Ilam. Vertical bar above each column represents standard deviation. Lowercase letters in the boxes indicate significant differences between declined and healthy trees from the two sites (Duncan’s multiple range test). The same letters indicate no significant differences (*p* < 0.05).

**Figure 2 plants-12-00377-f002:**
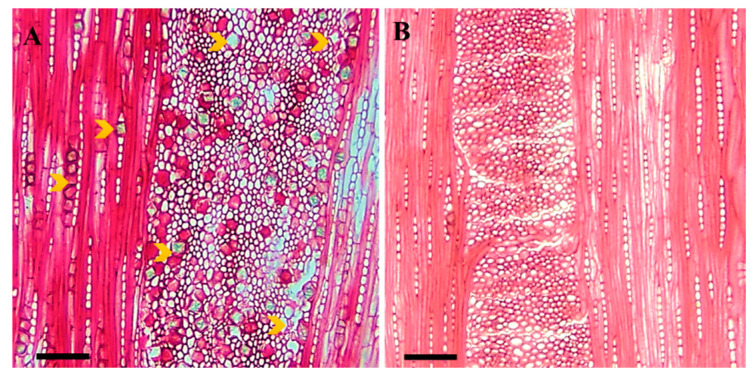
Frequency of crystals in the cells of the ray parenchyma of tangential sections in declined Persian oak wood from two different sampling sites: (**A**) Dareh-shahr, with high numbers, and (**B**) Ilam, with low numbers of crystals. Yellow arrows indicate crystals. Scale bar = 75 μm.

**Figure 3 plants-12-00377-f003:**
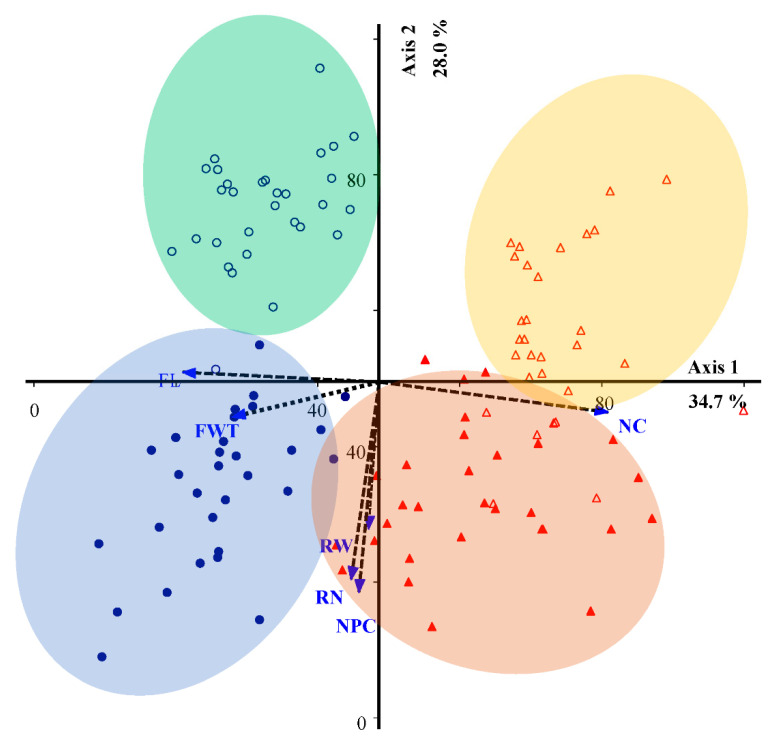
Principal component analysis (PCA) biplot based on correlation matrix of declined and healthy *Quercus brantii*. ▲ Declined trees from Dareh-shahr; △ healthy trees from Dareh-shahr; • declined trees from Ilam; ⚬ healthy trees from Ilam. Arrows with narrow angles from the axis indicate strong correlations, while a right angle indicates no correlation. The length of the arrow is a measure of the relative importance of the variables. FL, fiber length; FWT, fiber wall thickness; NPC, number of axial parenchymal cells; RN, ray number; RW, ray width; NC, number of crystals.

**Figure 4 plants-12-00377-f004:**
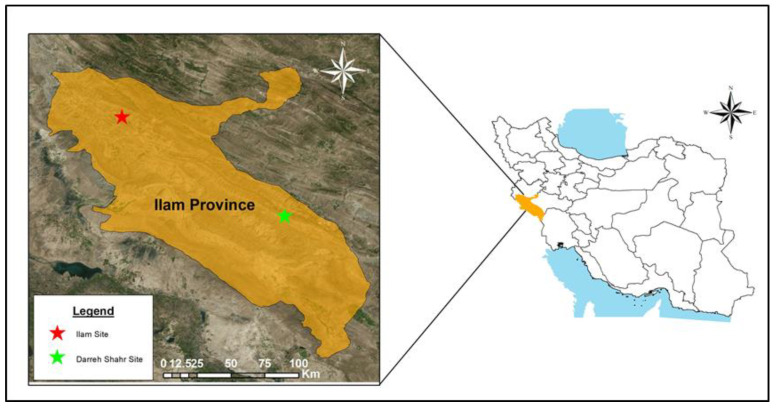
The locations of sampled forests stands on Ilam map, western Iran.

**Figure 5 plants-12-00377-f005:**
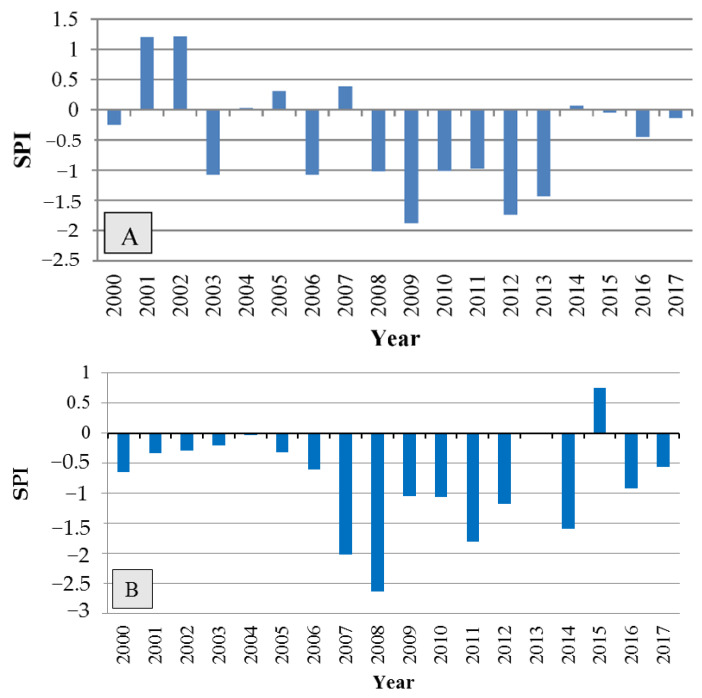
Standardized Precipitation Index (SPI) showing the severity and period of drought at the two sampling sites of Dareh-shahr (**A**) and Ilam (**B**).

**Table 1 plants-12-00377-t001:** Summary of two-way ANOVA of wood anatomical variables in sapwood of healthy and declined *Quercus brantii* affected by drought stress.

		Mean Square					
Source of Variation	df	FL (µm)	FWT (µm)	NPC (mm^−1^)	RN (mm^−1^)	RW (μm)	NC (mm^−2^)
Decline	1	0.206 ^ns^	37.899 **	113.733 **	142.474 **	21.338 **	0.500 ^ns^
Sites	1	67.690 **	43.954 **	0.308 ^ns^	2.882 ^ns^	1.125 ^ns^	135.380 **
Decline × Sites	1	11.384 **	4.686 *	0.308 ^ns^	18.871 **	15.213 **	0.30 ^ns^
Error	116	23,817,771	7.283	18.318	46.643	25.795	22.830

FL, fiber length; FWT, fiber wall thickness; NPC, number of axial parenchymal cells; RN, ray number; RW, ray width; NC, number of crystals. * The mean difference was significant at the 0.05 level. ** The mean difference was significant at the 0.01 level. ns, non-significant.

**Table 2 plants-12-00377-t002:** Wood anatomical traits used in the principal component analysis (PCA) on sapwood of healthy and declined *Quercus brantii* affected by drought stress.

Trait	Component	
	Axis 1	Axis 2
FL (µm)	−0.746 **	0.167 ^ns^
FWT (µm)	−0.644 **	−0.315 *
NPC (mm^−1^)	−0.237 ^ns^	0.770 **
RN (mm^−1^)	−0.280 ^ns^	−0.748 **
RW (µm)	−0.170 ^ns^	−0.643 **
NC (mm^−2^)	0.804 **	−0.291 ^ns^
Eigenvalues	2.431	1.962
% of variance	34.730	28.031

FL, fiber length; FWT, fiber wall thickness; NPC, number of axial parenchymal cells; RN, ray number; RW, ray width; NC, number of crystals. * The mean difference was significant at the 0.05 level. ** The mean difference was significant at the 0.01 level. ns, non-significant.

**Table 3 plants-12-00377-t003:** Main characteristics of the study sites and the precipitation and temperature data from the nearest meteorological station.

Forest Site	Altitude (m)	Longitude	Latitude	A.M.P (mm)	A.M.T (°C)	Max. T	Min. T	Aridity I.	Soil Type
Dareh-shahr (A)	933	33°3′30′′ N	47°19′30′′ E	465.1	19.5	44.7	2.6	Semi-arid (16.48)	Sand-loamy
Ilam (B)	1680	33°43′03′′ N	46°14′36′′ E	582.2	16.9	36.8	0.6	Mediterranean (21.65)	Loam-clay-sandy

A.M.P., Annual Mean Precipitation; A.M.T., Annual Mean Temperature; Max. T., Maximum Monthly Temperature; Min. T., Minimum Monthly Temperature; Aridity I.; De Martonne Aridity Index. The values inside the parentheses show the De Martonne Aridity Index. A.M.P and A.M.T are, respectively, the mean annual precipitations and temperatures for a period of 33 years from 1986 to 2018. Max. T and Min. T. are, respectively, the mean temperatures in the hottest and coldest months for a period of 33 years from 1986 to 2018 (Ilam Meteorological Bureau, 2018). Soil types were reported by Menati et al. [68].

## Data Availability

The datasets generated for this study are available at: 10.6084/m9.figshare.21438231 (accessed on 30 October 2022).

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
