# Peer review of "How Tree Decline Varies the Anatomical Features in Quercus brantii"

_plants, 2023, doi:10.3390/plants12020377_

Round 1
Reviewer 1 Report
The authors describ the influence of drought stress on variations in the anatomical characteristics of Quercus brantii wood was investigated. The paper brings interesting results related to plant defense against stress. These results may help to predict the responses of woody plants against climate change.
I think the manuscript is good written in all sections and the results contain important and significant findings that are suitable for the journal. I would like to suggest three points for further improvement
The term “cellular-anatomical structure” is related to cell ultrastructure. It is correct to use only anatomical structure
Why is figure A without scale? “Figure A is out of scale and scale bar in Figure B=75 μm.”
Figure B appears to be at a different magnification than Figure A, and so it is not possible to observe the difference in crystal distribution indicated in the figure.
Author Response
Dear Reviewer,
The comments were replied to point by point. TQ
The term “cellular-anatomical structure” is related to cell ultrastructure. It is correct to use only anatomical structure
*Was done…
Why is figure A without scale? “Figure A is out of scale and scale bar in Figure B=75 μm.”
*The Figure was corrected.
Figure B appears to be at a different magnification than Figure A, and so it is not possible to observe the difference in crystal distribution indicated in the figure.
*The Figure was corrected.

Reviewer 2 Report
The research is exciting and scientifically sound. However, I have some comments.
Abstract- Need to improve. Include results more specifically, add increase and decrease values in the abstract.
Introduction- Discuss about the importance of Oak plants. Discuss the novelty of the research work at the end of the introduction.
Result- The result section needs to significant change. Include values of all data in the result section.
Figure 1- Should represent as double panel. One under one is looking odd.
Figure 3- Mention the PCA values in the figure.
Line 193- Remove the color circle from the figure legend.
Reference- Check all the references.
Author Response
I herewith respond to the valued reviewer's comments and suggestions point-by-point.
Abstract- Need to improve. Include results more specifically, add increase and decrease values in the abstract.
*The mean values from the variables were inserted. RQ
Introduction- Discuss about the importance of Oak plants. Discuss the novelty of the research work at the end of the introduction.
*The importance of oak plants along with the Zagros forest was added.
P The novelty of the research work was explained at the end of the introduction. TQ.
Result- The result section needs to significant change. Include values of all data in the result section.
*The values were added.
Figure 1- Should represent as double panel. One under one is looking odd.
*I agree with the honorable referee's comment, although I prefer it to be done when formatting the pages.
Figure 3- Mention the PCA values in the figure.
*The values were added.
Line 193- Remove the color circle from the figure legend.
*Done. TQ
Reference- Check all the references.
*By inserting one new reference, all the references were rechecked.

Reviewer 3 Report
The reviewed manuscript raises very interesting issues concerning morphological changes and defensive capabilities of the Persian oak in conditions of prolonged drought.
Line 87: It would be worthwhile to clearly formulate the hypotheses, especially since the conclusions explain possible hypotheses.
Furthermore, the following sentence (lines: 256-258) –... 'The findings of this work provide a stronger base for establishing a functional hypothesis for the roles of parenchyma in wood, and changes in fiber and cellular crystals characteristics during the tree response to drought...' could also be an inspiration to make hypotheses.
Line 93: Figure 1 should be placed below Table 1, this arrangement makes it easier to follow the results
Line 134: Table 2 and Figure 3 should be next to each other
Line 350: Table 3 - The title and description of the table above the table, not below it
Line 398: What does it mean - ....were close to each other'. It would be worth specifying the distances between the trees.
Line 408: Nevertheless, it would be worthwhile to briefly provide information about sampling methods

Author Response
I herewith respond to the valued reviewers’ comments and suggestions point-by-point.
Comments and Suggestions for Authors
The reviewed manuscript raises very interesting issues concerning morphological changes and defensive capabilities of the Persian oak in conditions of prolonged drought.
Line 87: It would be worthwhile to clearly formulate the hypotheses, especially since the conclusions explain possible hypotheses.
*Done. TQ
Furthermore, the following sentence (lines: 256-258) –... 'The findings of this work provide a stronger base for establishing a functional hypothesis for the roles of parenchyma in wood, and changes in fiber and cellular crystals characteristics during the tree response to drought...' could also be an inspiration to make hypotheses.
*The new hypothesis was added in the last para of Intro as the valued reviewer asked. TQ
Line 93: Figure 1 should be placed below Table 1; this arrangement makes it easier to follow the results.
*Done. TQ
Line 134: Table 2 and Figure 3 should be next to each other
*Table 2 and Fig. 3 were arranged next to each other.
Line 350: Table 3 - The title and description of the table above the table, not below it
*Usually the title of the Table is above and the caption and descriptions come below the Table. I think that the respected reviewer made a mistake in Table 5 for Table 4.
Line 398: What does it mean - .... were close to each other. It would be worth specifying the distances between the trees.
*The sentence was modified.
Line 408: Nevertheless, it would be worthwhile to briefly provide information about sampling methods
*Some complementary info was added to the “Sampling Method” Section. At the middle and end of this section, the sampling methods were provided.

Round 2
Reviewer 2 Report
Accept in present form
Author Response
Manuscript ID: plants-2038059
Type of manuscript: Article
Title: How Crown Dieback Varies the Anatomical Features in Quercus brantii
Dear Editor,
Thank you very much for your email. The comments and suggestions are responded to point-by-point as followed:
* As my co-author, Prof. Woodward said, the question mark should be deleted in the Title. It was yellow-highlighted.
- I) Please revise your manuscript according to the referees’ comments and upload the revised file within 5 days.
*Done.
(II) Please use the version of your manuscript found at the above link for your revisions.
*Done.
(III) Please check that all references are relevant to the contents of the manuscript.
* Checked. There were some mistakes related to the References that were corrected. The changes were track-changed on the text.
(IV) Any revisions made to the manuscript should be marked up using the “Track Changes” function if you are using MS Word/LaTeX, such that changes can be easily viewed by the editors and reviewers.
* Done.
(V) Please provide a short cover letter detailing your changes for the editors’ and referees’ approval.
- Done.
Best Regards, Hamid
Hamid R. Naji
Corresponding Author (hrn_16hrn@yahoo.com)
+98-911-458-9774
